# A norovirus gastroenteritis outbreak in an Australian child-care center: A household-level analysis

**Nicolas Roydon Smoll**[1]*, **Arifuzzman Khan**[1], **Jacina Walker**[1,2], **Jamie McMahon**[3,4], **Michael Kirk**[5], **Gulam Khandaker**[1]

1 Central Queensland Public Health Unit, Rockhampton, Queensland, Australia, 2 Australian National University, Canberra, Australia, 3 Public Health Virology Laboratory, Forensic and Scientific Services, Brisbane, QLD, Australia, 4 Child Health Research Centre, The University of Queensland, South Brisbane, QLD, Australia, 5 Central Queensland Hospital and Health Service, Rockhampton, Queensland, Australia

* nrsmoll@gmail.com

**Data Availability Statement:** Data cannot be shared publicly because it contains sensitive personal data. Data are available from the ANU Ethics Committee for researchers who meet the

## Abstract

There is a large burden of norovirus disease in child-care centers in Australia and around the world. Despite the ubiquity of norovirus outbreaks in child-care centers, little is known about the extent of this burden within the child-care center and the surrounding household clusters. Therefore, we performed an in-depth analysis of a gastroenteritis outbreak to examine the patterns of transmissions, household attack rates and the basic reproduction number ($R_0$) for Norovirus in a child-care facility. We used data from parental interviews of suspected cases sent home with gastroenteritis at a child-care center between 24th of August and 18th of September 2020. A total of 52 persons in 19 household clusters were symptomatic in this outbreak investigation. Of all transmissions, 23 (46.9%) occurred in the child-care center, the rest occurring in households. We found a household attack rate of 36.5% (95% CI 27.3, 47.1%). Serial intervals were estimated as mean 2.5 ± SD1.45 days. The $R_0$, using time-dependent methods during the growth phase of the outbreak (days 2 to 8) was 2.4 (95% CI 1.50, 3.50). The count of affected persons of a child-care center norovirus outbreak is approximately double the count of the total symptomatic staff and attending children. In the study setting, each symptomatic child-care attendee likely infected one other child-care attendee or staff and just over one household contact on average.

## Introduction

Noroviruses are an important cause of acute gastroenteritis and implicated in almost a fifth of all episodes of gastroenteritis [1,2]. Worldwide, norovirus is one of the most common causes of diarrheal diseases [3]. The estimate of incidence of gastroenteritis in Australia is 17.2 million (95% CI: 14.5–19.9 million) cases per year [4]. Norovirus-related gastrointestinal illnesses, are one of the leading causes of acute gastroenteritis outbreaks in Australia, being implicated in over 50% of outbreaks in Australia, and therefore is often presumed to be the causative agent despite no confirmation [5]. In Australia, 2,046,184 lifetime productivity losses are observed due to norovirus infection [5].

criteria for access to confidential data. Contact is
human.ethics.officer@anu.edu.au.

**Funding:** The authors received no specific funding
for this work.

**Competing interests:** The authors have declared
that no competing interests exist.

Norovirus requires a close contact with others to spread, and thus is often associated with outbreaks in close-quarters situations such as child-care centers, aged care facilities and cruise ships. The basic reproductive number (the number of persons that one infected person will infect in an entirely susceptible population; $R_0$) of norovirus in various community settings internationally such as schools, colleges, and universities, is approximately 2.92 (95% CI 2.82–3.03), and in cruise ships this increases up to 7.2 [6.1, 9.5] [6,7]. Steele et al. 2020 found in a large scale review of 272 child-care center outbreaks a median attack rate of 21% (interquartile range [IQR] 13–36%), total involved 18 (11–29 persons) and an $R_0$ using the final size methodology of 2.67 (2.39–3.60) [6]. This work represents a detailed analysis of a child-care center outbreak using case interviews to better understand the extent of the outbreak into households which most often goes unreported.

Norovirus gastroenteritis is not a notifiable disease in Australia [8]. There is a plethora of information regarding the microbiology, genomics and global epidemiology of norovirus [6,9–16]. But information on how a simple norovirus outbreak affects the child-care center and households associated with the child-care center is scarce. An outbreak of norovirus infection occurred in an Australian child-care center in August and September of 2020 provided us with an opportunity to perform an in-depth review of the patterns of transmissions, household attack rates and the basic reproduction number ($R_0$) for Norovirus in child-care facilities. In addition, this provides insights into the routes of COVID-19 transmission within a child-care center.

## Materials and methods

We used data from contact tracing interviews with parents of suspected cases sent home with acute gastroenteritis of unknown cause at a child-care center between 24th of August and 18th of September in 2020. All parents provided oral informed consent, and interviews conducted within the laws set out in the Public Health Act of 2005. The child-care center is licensed to care for 103 individuals, but an average of 68 infants and children attend the facility on a particular day during the outbreak. A detailed review of the history revealed that this was likely a norovirus outbreak. A outbreak of norovirus is defined when there is greater than 50% of infected individuals that have vomiting and greater than 50% with diarrhea [17,18]. In addition, the Centers for Disease Control in the USA define a norovirus outbreak as two or more similar illnesses resulting from a common exposure that is either suspected or laboratory-confirmed to be caused by norovirus [19]. A suspected case of gastroenteritis was defined as a person whom had 2 or more episodes of vomiting and/or 3 or more episodes of diarrhea within last 24 hours [20]. A confirmed case was a person whom provided a sample that tested positive for norovirus in the faeces by real-time polymerase chain reaction (PCR). Samples from suspected norovirus cases were sent to the Public Health Virology laboratory, Forensic and Scientific Services, Queensland, Australia. A modified real-time reverse transcription PCR capable of detecting and differentiating norovirus genogroups GI and GII was utilized based on published primers and probes targeting ORF1–ORF2 junction region [21]. Sensitivity and specificity by in-house validation were deemed 100% and 99.06% respectively.

Children are those aged under ten years of age, and infants are those aged less than 1 year of age. Measures to control the outbreak began the day we were alerted to the outbreak (31st of August) e.g. regular cleaning with a bleach solution, removing toys, alerting parents and keeping unwell children at home.

We performed case interviews to understand the household attack rates and routes of transmission across 18 household clusters. The child-care center manager provided daily lists of suspected cases that are not attending or have been sent home with symptoms of

gastrointestinal illness. A case interview was performed by a doctor (NRS) with the parents responsible for each child sent home due to symptoms of gastroenteritis (vomiting and/or diarrhea), within 2 weeks of the illness. Information on family members affected, their symptoms (must meet case definitions) and date of symptom onset, household size and likely route of transmission was collected, and ensured that other gastrointestinal complaints were not the cause of the symptoms. Onset times, age, early care room assignment, family clusters and symptoms were also collected. Verbal consent was obtained by parents or guardians as interviews were conducted over the phone. This investigation was approved (including the use of oral consent) by the Australian National University Human Research Ethics Committee (HREC Ref number 2020/629).

We aimed to ensure that each family cluster had a rational transmission route consistent with gastroenteritis. Infector/infectee clusters were created by identifying cases that frequented the early care center and introduced the infection to the family cluster. The child that brought the infection home was paired with each family member to identify infector/infectee pairs used for the calculation of the serial interval. The serial interval was the time between the onset of symptoms for each infector/infectee pair, and calculated the generation time from the interval [22]. The first generation infection is obtained by the child at the child-care center and the second generation infection is all subsequent household infections.

The child-care center attack rate was the number of attendee cases divided by the overall average attendees (excludes staff). The household attack rate is the number of household cases (excluding the infant/child who acquired the infection at the child-care center) divided by the total persons in the household (excluding the infant/child who acquired the infection at the child-care center). Our approach to modelling the basic reproduction number for this outbreak was performed using a time-dependent estimation of $R_0$, and all child-care staff, attendees and household members assumed to be susceptible to norovirus [23,24]. We chose the $R_0$ estimated over day 2–8 as the $R_0$ for this outbreak. Reproductive numbers less than one suggest that control of the outbreak has been achieved as each case (on average) is only passing it on to less than one other person.

The household $R_0$ was calculated using hand counting and derived from the overall attack rate. Obtaining the household $R_0$ by hand counting consists of the average household transmissions (second generation infections) occurred in each household cluster. Note that the overall $R_0$ could not be calculated from hand counting as we could not identify links between the children inside the child-care center. The attack rate method is as follows:

$$Household\ R_0\ =\ -\frac{log(1-AR)}{AR}$$

## Results

We used data from parental interviews of suspected cases sent or kept home with gastroenteritis at a child-care center between 24[th] of August and 18[th] of September 2020, with the authors becoming aware of the outbreak on the 31[st] of August. A total of 17 children and two staff members (19 household clusters) were symptomatic and met the criteria for suspect or confirmed norovirus during this outbreak investigation. A child-care center with an average of 68 children attendees per day reported an outbreak seven days after the primary case vomited in the nursery room, prompting an outbreak where a total of 52 persons, including 4 infants, 24 children, 22 non-staff adults and 2 staff members met the suspect (n = 51) or confirmed case (n = 1) definition. The parents of three children (n = 3/52) were not contactable. The total exposed during this outbreak was approximately 149 persons (see Table 1 for details). Seventeen children in six rooms of the child-care center were affected for a child-care center

**Table 1. Outbreak descriptors.**

| Total Infections, n = 52 (%) | |
|---|---|
| Infants | 4 (7.7) |
| Children | 24 (46.1) |
| Household Adults | 22 (42.3) |
| Staff | 2 (3.9) |
| Vomiting, n (%) | 42 (80.7%) |
| Diarrhoea, n (%) | 36 (69.2%) |
| Exposed | |
| Total HH Affected | 19 |
| HH Occupants, median (range) | 4 (2, 6) |
| Total Average Child-care Attendees[a] | 68 |
| Total Staff | 24 |
| Total HH Exposed[b] | 74 |
| Total Exposed | 149 |
| Medical Outcomes | |
| Hospitalized, n (%) | 2 (4.3) |
| Deaths | 0 |
| Attack Rates, (95% CI) | |
| Child-care Center Attendee | 25% (16.2, 36.4%) |
| Household[c] | 36.5% (27.3, 47.1%) |
| Staff | 8.3% (2.3, 25.8%) |
| Overall | 34.9% (27.7, 42.8%) |
| Basic Reproductive Numbers, (95% CI) | |
| HH (hand counted) | 2 (1.48, 2.51) |
| HH Attack Rate Derived $R_0$ | 1.24 |
| Overall Attack Rate Derived $R_0$ | 1.23 |
| Overall Time-dependent $R_0$ (Day 2–8) | 2.4 (1.50, 3.50) |

[a]Average number includes a small number of HH exposed but unaffected persons. This results in a slight underestimation of the attack rates due to the potential double counting of unexposed siblings that attended the center but were unaffected.

[b]Infected children, infants and staff (n = 19) that attend the child-care center are included here and also included in the total average child-care center attendees.

[c]Total exposed in the household was 55, or the total exposed minus the child-care center cases (n = 19). Total household transmissions was 26.

Total staff represents the total number of staff with potential contact with child-care center attendees. The overall attack rate derived $R_0$ is different from the time-dependent method as it covers the entire outbreak, before and after mitigation strategies, which is why the time-dependent method is considered the optimal method used to define the basic reproductive number of norovirus outbreaks in child-care centers. HH = households.

attendee attack rate of 25% (95% CI 16.2, 36.4%). Two of the 17 children, both infant(s) were hospitalized for observation only. Vomiting occurred in 42 (80.8%) and diarrhea occurred in 36 (69.2%) of cases. Of all transmissions, 23 (46.9%) occurred in the child-care center, the rest occurring in households (see Figs 1 and 2). Cluster 8 (3 persons) were excluded from the serial interval estimates because of unclear transmission pathways. We were only able to obtain a single stool specimen and confirm a single case (Norovirus Genus II) with PCR as parents often disregarded requests to obtain a specimen as the disease was transient and, in some cases, mild.

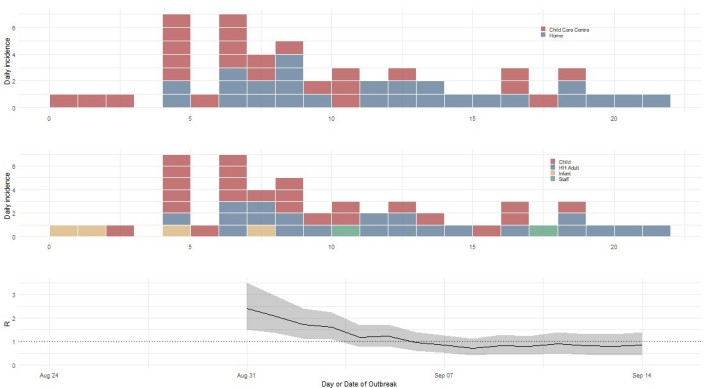

**Fig 1. Epicurves and time-dependant $R_0$ of cases by likely transmission location.** Demonstrates a large burden of disease acquired within households (top graph), with 23 (46.9%) of transmissions occurring in the child-care center, the rest occurring in households. The middle graph demonstrates the burden of disease at the child-care center and at associated households. Each block represents one case and "HH Adult" represents an adult in the household. R is a sliding window estimate which is why there are no estimates for the first 7 days. One case (staff member) could not recall onset dates and not shown here.

Case interviews revealed that the primary case for this outbreak was an infant who probably transmitted infection via multiple unspecified intermediaries (other infants or children) to 19 the infant/child cases (and subsequently 26 more were infected) in this outbreak (Fig 2). The primary case was often dropped off at the center early in the morning where this child would mix with other children of varying ages, ensuring transmission across rooms in the center. The primary case had no other siblings that attended the center.

Overall, there was 55 persons within households that were exposed, and 23 infected, for a household attack rate of 36.5% (95% CI 27.3, 47.1%). The household $R_0$ was between 1.24 and

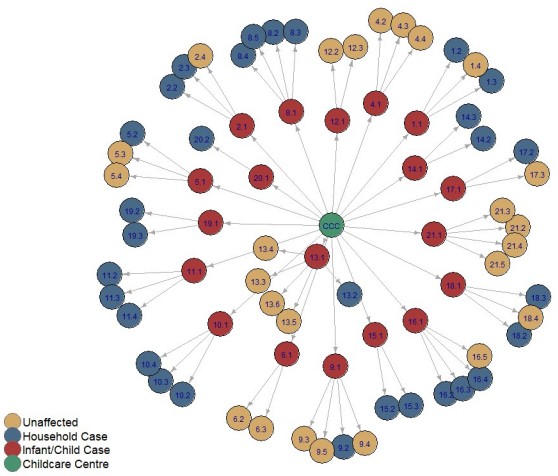

**Fig 2. Network model of the outbreak.** The index cases represent the children that obtained the infection at the child-care center and introduced Norovirus to the household. The red circles represent the child or staff member that was infected at the child-care center and introduced the infection to a single family cluster (blue for household suspected cases, and beige for household members that did not acquire the infection). The third row represents households of children infected at the center. The numbers within the circles represent the cluster ID (first number) and family member ID (second number). The parents of three children were not contactable. The cause of the high variability of individual household attack rates (e.g. some with no transmission and some with 100% transmission) remains unclear.

2 (see Table 1). Note that the household $R_0$ cannot be greater than 4 as very few households had 4 or more total people in them.

On closer inspection, there were 19 contactable infants/children or staff that acquired the infection at the child-care center and brought the infection to the family cluster, resulting in 26 infector/infectee pairs. The resultant serial intervals (mean 2.5 days, SD 1.45) were modelled establish a reasonable generation time distribution for estimation of the reproductive number ($R_0$) and found a gamma distribution fit best on our serial intervals resulting in a generation time distribution with a mean of 2.6 and SD 1.35. The $R_0$, using time-dependent methods during the growth phase of the outbreak (days 2 to 8) was 2.4 (95% CI 1.50, 3.50). and control was likely achieved sometime between day 10–16 when the lower 95% CI of R crossed 1 at day 12 (five days after the commencement of control measures) and mean R crossing 1 on day 14. The upper 95% CI did not cross one.

## Discussion

This detailed review of a child-care center Norovirus gastroenteritis outbreak demonstrated a basic reproduction number of 2.4 (95% CI1.50, 3.50), and that household members of a child with norovirus in a child-care facility probably has a greater than 35% chance (attack rate of 36.5% [27.3, 47.1%]) of being symptomatic with norovirus, i.e. each affected child is expected to infect just over one person at home (household $R_0$ = 1.23). However, the generalizability of these estimates to other areas, as infection-control strategies may have started earlier or later, and different regions have differing household sizes and compositions.

Despite the current COVID-19 restrictions and heightened infection control measures, such a high attack rate observed during these outbreaks have real life implications. During an outbreak of norovirus in a child-care facility, for each child and staff member that gets affected, there is at least one person affected in a household related to the child-care center, indicating that the size of an outbreak is likely ~2x the size of what is seen at the child-care center (only 48% of transmissions occurred in the child-care center). We suspect that variable adherence to infection control practices can lead to high attack rates and attempted to identify transmission routes that have implications for a COVID-19 outbreak in a child-care center.

We found three routes of transmission across rooms in a child-care center. Firstly, there is a single unified room where children will mix and transmit if they are dropped off early in the morning or kept late in the afternoon. Secondly, if children have an older or younger sibling that spends their day in a different room, the transmission route to the other room may occur via the sibling. Thirdly, staff work across several rooms and may cross infect. Overall, cohorting staff and children by avoiding mixing, and keeping siblings of unwell children at home should help cut the chains of transmission.

Detailed risk factors for gastroenteritis outbreaks and prevalence of Norovirus in child-care centers was explored by Ensrink et al. 2015 in the Netherlands [25]. The risk factors included: large capacity of the child-care center, crowding, having animals, nappy changing areas, sandpits, paddling pools, cleaning potties in normal sinks, cleaning vomit with paper towels (but without cleaner), mixing of staff between child groups, and staff members with multiple daily duties. Protective factors included: disinfecting fomites with chlorine, cleaning vomit with paper towels (and cleaner), daily cleaning of bed linen/toys, cohorting and exclusion policies for ill children and staff.

The limitations of this analysis include that we modelled only a single outbreak which had one PCR confirmed case. Our modelling was limited to a simple model of infection with our infector/infectee pairs having assumed links based on the household location of the child. While this may appear overly simplistic, in most interviews it appeared to be reasonable

(except cluster 8 which was excluded). While the case interview method is good for identifying household cases, we did not call all families of the center and there may be others that were unwell but did not report to the center. Children with mild, or subclinical symptoms may also have been missed. The strengths included a detailed case analysis with robust information on onset dates, household sizes, likely infector/infectee pair and symptom analyses.

## Conclusion

A child-care center norovirus outbreak is approximately double the size of the total symptomatic staff and attending infants/children. In the study setting, each symptomatic child-care attendee likely infected one other child-care attendee or staff and just over one household contact on average. The practice of mixing children during morning drop-off and late afternoon pickup as well as keeping the well sibling(s) of an affected child at home are likely valuable strategies to mitigate the further transmission. Further similar analyses are warranted to better understand the variability of the presented estimates.

## Author Contributions

**Conceptualization:** Nicolas Roydon Smoll, Arifuzzman Khan, Gulam Khandaker.

**Data curation:** Nicolas Roydon Smoll, Arifuzzman Khan.

**Formal analysis:** Nicolas Roydon Smoll, Arifuzzman Khan.

**Investigation:** Nicolas Roydon Smoll, Jacina Walker, Jamie McMahon.

**Methodology:** Nicolas Roydon Smoll, Jacina Walker, Jamie McMahon, Gulam Khandaker.

**Project administration:** Michael Kirk.

**Resources:** Michael Kirk.

**Supervision:** Gulam Khandaker.

**Validation:** Jamie McMahon.

**Visualization:** Nicolas Roydon Smoll.

**Writing – original draft:** Nicolas Roydon Smoll, Gulam Khandaker.

**Writing – review & editing:** Nicolas Roydon Smoll, Arifuzzman Khan, Jacina Walker, Jamie McMahon, Michael Kirk, Gulam Khandaker.

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
