## [Decision Letter · Decision Letter 0]

26 May 2021

PONE-D-21-06174

Extent of a norovirus gastroenteritis outbreak in a childcare centre:  a household-level review.

PLOS ONE

Dear Dr. Smoll,

Thank you for submitting your manuscript to PLOS ONE. After careful consideration, we feel that it has merit but does not fully meet PLOS ONE’s publication criteria as it currently stands. Therefore, we invite you to submit a revised version of the manuscript that addresses the points raised during the review process.

The Authors are expected to address all the criticisms by all Reviewers. In particular, please clarify the case definition clearly (Reviewers #1 and #2), provide background information on the study population and setting (Reviewer #1), clarify distinction between norovirus and gastroenteritis, case ascertainment, number of samples collected and sample selection, and improve Figure 2 (Reviewer #2). In additional to the above comments, please address,

The title can be improved as something like “A norovirus gastroenteritis outbreak in an Australian childcare centre: a household-level analysis”Title. Please reconsider the use of ‘norovirus gastroenteritis’ as only 1 child was confirmed with norovirus.Abstract, “Serial intervals were estimated as 2.5± 1.45 days”. Please clarify whether 1.45 is a SD or SE. Also, do you mean the “mean serial interval”?L78. Please provide the missing reference “[REF]”L136. Please provide a reference for the statement “generation time distribution with a mean of 2.6 136 and SD 1.35”Figure 2 can be improved by providing the types of cases, e.g. children, staff, household member etc.

We look forward to receiving your revised manuscript.

Kind regards,

Eric HY Lau, Ph.D.

Academic Editor

PLOS ONE

Journal Requirements:

2. In the Methods, please clarify that participants provided oral consent. Please also state in the Methods:

- Why written consent could not be obtained

- Whether the Institutional Review Board (IRB) approved use of oral consent

- How oral consent was documented

- Whether consent was informed

For more information, please see our guidelines for human subjects research: https://journals.plos.org/plosone/s/submission-guidelines#loc-human-subjects-research

3.We note that you have indicated that data from this study are available upon request. PLOS only allows data to be available upon request if there are legal or ethical restrictions on sharing data publicly. For information on unacceptable data access restrictions, please see http://journals.plos.org/plosone/s/data-availability#loc-unacceptable-data-access-restrictions.

Additional Editor Comments:

The Authors are expected to address all the criticisms by all Reviewers. In particular, please clarify the case definition clearly (Reviewers #1 and #2), provide background information on the study population and setting (Reviewer #1), clarify distinction between norovirus and gastroenteritis, case ascertainment, number of samples collected and sample selection, and improve Figure 2 (Reviewer #2). In additional to the above comments, please address,

1. The title can be improved as something like “A norovirus gastroenteritis outbreak in an Australian childcare centre: a household-level analysis”

2. Title. Please reconsider the use of ‘norovirus gastroenteritis’ as only 1 child was confirmed with norovirus.

3. Abstract, “Serial intervals were estimated as 2.5± 1.45 days”. Please clarify whether 1.45 is a SD or SE. Also, do you mean the “mean serial interval”?

4. L78. Please provide the missing reference “[REF]”

5. L136. Please provide a reference for the statement “generation time distribution with a mean of 2.6 136 and SD 1.35”

6. Figure 2 can be improved by providing the types of cases, e.g. children, staff, household member etc.

Reviewers' comments:

Reviewer's Responses to Questions

**Comments to the Author**

1. Is the manuscript technically sound, and do the data support the conclusions?

Reviewer #1: Partly

Reviewer #2: Partly

2. Has the statistical analysis been performed appropriately and rigorously? 

Reviewer #1: Yes

Reviewer #2: I Don't Know

3. Have the authors made all data underlying the findings in their manuscript fully available?

Reviewer #1: Yes

Reviewer #2: Yes

4. Is the manuscript presented in an intelligible fashion and written in standard English?

Reviewer #1: Yes

Reviewer #2: No

5. Review Comments to the Author

Reviewer #1: Overall comments

The present study describe the patterns of transmissions, household attack rates and the basic reproduction number (R0) for and small outbreak of Norovirus occurred in Australia between August and september in 2020.

Overall, the study is appropriate to answer the aim proposed regarding to clarify the effects of Norovirus infecction in a childcare centre on the household transmition.

Despite of some significant limitations the results could be used as a reference to also help in identifying new control strategies for future epidemics.

I am grateful to the authors for the opportunity to review their preprint and hope my comments may be useful.

Overall strengths and impact

The abstract is written succinctly and clearly.

The research question is clear and could have important implications for Norovirus surveillance.

Specific comments on weaknesses

Major points

1. In the Materials and Methods section, in my opinion, it will be more informative if the authors describe clearly some aspects, e.g.:

- The staff of the child care centre, the number of the children associated to the same, their age range. As the same, it will be fruitful if the authors also describe the number of household studied, maybe the median of the number of the persons in each one. Some of this information is mentioned in Materials and Methods but missing in the results as well as some is given as a results, but I consider that is the main data from which the authors started the analysis.

- Since only one case was confirmed by PCR, it could be useful if the authors reinforce the criteria to consider one case as Norovirus positive, maybe others clinical symptoms and signs with any epidemiological relationship. It will be informative if they also add who (laboratory) and how (methodology) it was performed the confirmatory PCR.

2. In the Discussion section:

- I consider that the authors could improve the discussion if they compare as much as the literature it allowed, to any similar study, even not necessary Australian.

Minor points

1. Please delete the repeated word “that” in line 29.

2. Please insert the reference in line 78.

3. In Figure 2, I advise to change the colors corresponding to unaffected and child care center circles to avoid any confusion with Figure 1 colors. I also recommend to improve the quality of the image, please consider to reproduce with another tools since it could be possible that the program not allowed.

4. I consider that it will be informative if the authors declare what they consider as children and infant, maybe in terms of age.

5. In my opinion, It will be more informative if the authors apply the primary or index case concept to the first case and secondary cases to the subsequently cases, and so on.

Reviewer #2: Review of PONE-D-21-06174: "Extent of a norovirus gastroenteritis outbreak in a childcare centre: a household-level review," by Smoll et al.

Reviewer comments (by section of manuscript):

General: PLOS ONE "does not copyedit accepted manuscripts," and so this reviewer has endeavored to identify text that might be altered in the interest of being "clear, correct, and unambiguous."

Title: Consider adding "presumed" or "probable" before "norovirus."

Abstract:

Several of this reviewer's comments on the body of the paper (see below) should be incorporated into the abstract. In particular, the distinction between norovirus and the broader category of gastroenteritis is quite important. Also, the term "affected persons" probably means "symptomatic household contacts of sick childcare attendees (and/or staff???)" but readers who have not yet read the full paper may not not interpret the term "affected" in that way. In the summary statements, it might be clearer to say that "in the study setting, each symptomatic childcare attendee apparently infected approximately two household contacts, on average."

Ethics statements:

The study was approved by the Australian National University Human Research Ethics Committee and the reference number is provided. Parents or guardians gave consent (may we assume that they consented both for their children and themselves?). Restrictions on data-sharing are meant to protect participants' confidentiality. No concerns.

Introduction:

This paper apparently describes the first investigation ever conducted of a presumed norovirus outbreak in childcare centers in Australia. Thus, it is of considerable public-health importance in the region, and may help set the standard for future surveillance.

Early in this section, it would be important to discuss the distinction between acute gastroenteritis and presumed or confirmed norovirus. Here or in "materials and methods" one could cite standard case definitions for "presumed norovirus," e.g. the Kaplan and/or Lively criteria (Kaplan J E, Ann Int Med 1982 June;96(6 pt 1):756-61. Lively J Y, Open Forum Infection Diseases 5(4): April 2018.).

The statement that there is only "minimal information available regarding the R0 in childcare facilities" may only refer to the present study setting. For example, Steele et al. (Steele, MK et al. Characterizing norovirus transmission from outbreak data, United States. Emerging Infectious Diseases 2020; 26(8):1818-1825) evaluated norovirus data from 272 childcare facilities in the United States.

The basic biologic description of norovirus might be shortened.

To smooth the transition to the "materials and methods" section, the introduction might include a basic statement about the outbreak that was investigated - e.g. "An outbreak of suspected norovirus infection occurred in an Australian childcare center in August and September of 2020. We took advantage of this outbreak to provide a view......."

Materials and Methods:

In the first paragraph of the section, the paper offers a suspected case definition for gastroenteritis but none for norovirus. This is confusing, because the topic of interest (based on the paper's title) is norovirus, which must be distinguished from the broader category of "gastroenteritis." A case definition of "presumed norovirus infection" is needed, as noted above (criteria of Kaplan or Lively). A definition of "confirmed" norovirus is given, appropriately, based on fecal PCR testing.

There should be a statement about the likely completeness of case ascertainment; for example, might children with only mild symptoms have been missed?

Was there a specific procedure for identification of "parents of suspected cases?" And was there a standard interview guide for these parents? In particular, it would be helpful to know whether the interview was designed to distinguish systematically between probable norovirus and other types of gastrointestinal complaints.

With regard to the identification of "rational transmission route," the concern about time course is necessary. But it would also be important to confirm that both childcare attendees and other household members had symptoms that met the norovirus case definition.

Here or in the results section, it would be informative to state the number of stool specimens tested, the lag time between symptom onset and stool testing, and who was tested (symptomatic children, staff, household contacts; others). If many stool specimens were collected but failed to detect norovirus, the likely reasons for this pattern should be mentioned in the discussion section.

With regard to estimation of R0: Were all child-care staff, attendees, and household members presumed to be susceptible to norovirus at the outset of the outbreak?

Lines 74 - 79: The term "infector/infectee pair" doesn't seem quite right, given that several sick children were linked to multiple infectees (up to 3 in a single household per Figure 2). Perhaps "infector/infectee cluster" would work.

Lines 80-81 (and 92): More justification should be given for using the denominator of “overall average attendees” instead of “total number of potentially exposed attendees.” Based on the data given, it seems that children who attended for some but not all days during the high-transmission interval might still have been at considerable risk of infection. Also, children who did not attend the childcare center every day probably do not comprise a random sample of the other attendees, and might be systematically different in ways that could affect risk of acquisition of norovirus. Finally, might some of the absent children have stayed home because of norovirus symptoms?

In lines 86 - 87: See comments re Figure 1.

Results:

A basic description of the study population might aid the reader. This could be presented in table form. It might describe total numbers of potentially exposed persons, not just numbers of persons who were known to be symptomatic, and categorize them by age (adult vs. child), childcare center vs. household status, and symptom status (in particular, whether symptoms satisfied the definition of presumptive norovirus). This suggestion is made because, at present, some information in the text is challenging to follow. For example, the reader is told that 47 persons in 18 households were "affected" but the text does not state that the 47 persons were apparently a mixture of childcare attendees and others, and the reader is not given the total population at risk.

A simple chronological description of the outbreak might also help the reader follow the narrative more easily. It could begin by stating - as the abstract does - that one childcare attendee had an episode of vomiting in August 2020 (was it August 24th?); that between 8/24/2020 and 9/18/2020, _____(number/%) of the attendees developed gastroenteritis symptoms and were sent home; and that on 8/31, the authors were informed of the outbreak, and efforts to control the outbreak were begun. This information already appears elsewhere but it would be useful to reiterate it in this section.

Lines 92-102: This paragraph is hard to follow. For example, the relationships between the “average of 68 children” (line 92), the 3 infants and 22 children (line 95), the 16 children in line 96, and the 2 infants of line 97 are unlikely to be clear to the reader at first glance, and most of these numbers seem to denote numerators (of cases) but not denominators (of exposed population). In the same section, it might be wise to say "developed symptoms that met the case definition for presumed norovirus," instead of “were affected.”

Lines 96-98: Were the two hospitalized infants a subset of the sixteen symptomatic children? Or were there eighteen symptomatic children?

Lines 104-111: Is this section meant to be the legend for Figure 1? This text might flow better as a figure legend than as a paragraph in the body of the paper. Similar query re lines 113-120.

Line 123: “the index” vs. “the index child”

Line 124: “they mixed” vs. “this child mixed”

Line 127: Important to clarify the identity and subgroup composition of the 67 persons. If attack rate for households was 82.4% of 67 persons, total number of symptomatic persons should have been 55 (67 x 0.824). But per Figure 2, the total number of symptomatic childhood attendees and symptomatic household residents was only 44 (18 primary cases, 26 secondary cases), which would be 65.7% of the 67. Does the apparent discrepancy have something to do with the “excluded” cluster or clusters?

Discussion:

Line 142 and subsequent paragraphs: See above re gastroenteritis vs. presumed or confirmed norovirus. The discussion should give the reader a data-based impression of the likelihood that this outbreak was caused by norovirus. The authors are entirely correct that norovirus is high on the list of etiologic possibilities, but the paper’s conclusion should be substantiated.

Lines 144-5. See above re “attack rate of 82.4%.”

Lines 147-155: Can you describe the Covid-19 mitigation strategies that were already in use at the childcare center before the gastroenteritis struck, especially those that might have had an impact on norovirus transmission? Also consider introducing the Covid-19 theme earlier in the paper as well as here.

Lines 149-152: Seems to conflict with previous observation that small household sizes limited possible magnitude of transmission. It would also be interesting (perhaps in future studies) to describe household-level strategies that might have prevented transmission, given that multiple households apparently had zero symptomatic transmission.

Lines 156-162: Very, very helpful and important.

Lines 163 – 170: Also very helpful and important. Some of this information might fit better in the introductory sections of the paper, though.

Lines 171-2: Again, more information is needed re number of stool specimens collected and possible explanations for absence of more confirmed norovirus specimens.

Conclusion:

Lines 179-181: If the focus of the work really is on norovirus, and if the sick children and their sick household contacts had symptoms that satisfied the definition of presumed norovirus, it might be preferable to state that “in the outbreak under investigation, each child with presumed norovirus probably infected at least two other household members (on average).” However, based on what has been presented in the paper, it seems premature to make predictions about the generalizability of this estimate of R0 to other sites with different household compositions and/or different infection-control strategies (for example, if infection-control strategies had not been implemented when the outbreak investigation began, or if household sizes had been larger, far more people might have become symptomatic and the Ro would likely have risen).

Lines 181-4: These recommendations and conclusions seem sound.

Figure 1:

This graphic gives the reader a very clear view of the evolution of the outbreak. It may also be the only mention of the very low number of infected childcare center staff. One drawback is that some of the red boxes denote children who attend the childcare center but other identical red boxes denote children who apparently did not attend; this is somewhat confusing but could be fixed easily.

It would be very helpful to see additional R0 estimates for the 1st week (Aug 24-31). Based on the visual evidence (17 cases in 7 days per the invaluable Figure 1), and the serial interval of approximately 2.5, R0 for the first week should be substantially higher than for subsequent weeks (after mitigation began).

Figure 2:

This is also a very appealing graphic, and it is fascinating to see the heterogeneity of apparent infectivity across households -- no secondary transmission at all in about 4 households, but 100% transmission in 4 other households; the reader may be quite interested to know if this is just chance or if something else is in play.

That said, Figure 2 might benefit from two modifications. First, the lines connecting different infection pathways sometimes overlap; the overlaps affect five households. The pathways for different households should be separated visually. Second, it would be helpful to the reader if the "unaffected" persons were described more fully. For example, are all of the "unaffected" persons adult members of an affected child's household, or are some of them asymptomatic childcare attendees?

Summary:

This paper should make a very useful contribution to public health in Australia, and other similar settings. The most essential building blocks of outbreak investigation are included (the main exception being laboratory confirmation). The introduction suggests that the work described might have been Australia's first norovirus outbreak investigation in a childcare facility, and the authors should therefore be commended for stepping up so enthusiastically. The paper would benefit from improvements in data presentation (for completeness and to facilitate reader comprehension) and from more systematic definition and use of the terms "gastroenteritis" and "norovirus," in the interest of making the importance and implications of the work even clearer to the reader.

6. PLOS authors have the option to publish the peer review history of their article (what does this mean?). If published, this will include your full peer review and any attached files.

Reviewer #1: **Yes: **Magilé C Fonseca

Reviewer #2: No

---

## [Author Response · Author response to Decision Letter 0]

9 Jul 2021

In the Methods, please clarify that participants provided oral consent. Please also state in the Methods:

- Why written consent could not be obtained

- Whether the Institutional Review Board (IRB) approved use of oral consent

- How oral consent was documented

- Whether consent was informed

All interviews were over the phone. “All parents provided oral informed consent, and interviews conducted within the laws set out in the Public Health Act of 2005”

We note that you have indicated that data from this study are available upon request. PLOS only allows data to be available upon request if there are legal or ethical restrictions on sharing data publicly. For information on unacceptable data access restrictions, please see

If there are ethical or legal restrictions on sharing a de-identified data set, please explain them in detail (e.g., data contain potentially identifying or sensitive patient information) and who has imposed them (e.g., an ethics committee). Please also provide contact information for a data access committee, ethics committee, or other institutional body to which data requests may be sent.

If there are no restrictions, please upload the minimal anonymized data set necessary to replicate your study findings as either Supporting Information files or to a stable, public repository and provide us with the relevant URLs, DOIs, or accession numbers. Please see http://www.bmj.com/content/340/bmj.c181.long for guidelines on how to de-identify and prepare clinical data for publication. For a list of acceptable repositories, please see http://journals.plos.org/plosone/s/data-availability#loc-recommended-repositories.

Minimum dataset will be uploaded

The Authors are expected to address all the criticisms by all Reviewers. In particular, please clarify the case definition clearly (Reviewers #1 and #2), provide background information on the study population and setting (Reviewer #1), clarify distinction between norovirus and gastroenteritis, case ascertainment, number of samples collected and sample selection, and improve Figure 2 (Reviewer #2). In additional to the above comments, please address,

1. The title can be improved as something like “A norovirus gastroenteritis outbreak in an Australian childcare centre: a household-level analysis”

The title has been has been changed as suggested. 

2. Title. Please reconsider the use of ‘norovirus gastroenteritis’ as only 1 child was confirmed with norovirus.

Thanks for this helpful comment. The authors have discussed this at length, and feel that the outbreak meets the criteria for a norovirus outbreak. See “Centers for Disease Control and Prevention. Responding to Norovirus Outbreaks. Published April 5, 2021. Accessed June 25, 2021. https://www.cdc.gov/norovirus/trends-outbreaks/responding.html “

3. Abstract, “Serial intervals were estimated as 2.5± 1.45 days”. Please clarify whether 1.45 is a SD or SE. Also, do you mean the “mean serial interval”?

The sentence was updated to “Serial intervals were estimated as mean 2.5 ± SD1.45 days”

4. L78. Please provide the missing reference “[REF]”

Apologies, that was an unintentional inclusion. We do not have a reference for a “two generation” model, it is simply what we did. There are no other childcare centre outbreaks which we can reference.

5. L136. Please provide a reference for the statement “generation time distribution with a mean of 2.6 136 and SD 1.35”

Added the reference to the methods section where we calculated the generation time distribution from the serial interval.

6. Figure 2 can be improved by providing the types of cases, e.g. children, staff, household member etc.

This is an appropriate suggestion and clarifies the presented data. The figure legend has been updated to provide this information.

Specific comments on weaknesses

Major points

1. In the Materials and Methods section, in my opinion, it will be more informative if the authors describe clearly some aspects, e.g.:

- The staff of the child care centre, the number of the children associated to the same, their age range. As the same, it will be fruitful if the authors also describe the number of household studied, maybe the median of the number of the persons in each one. Some of this information is mentioned in Materials and Methods but missing in the results as well as some is given as a results, but I consider that is the main data from which the authors started the analysis.

This is a thoughtful comment improving the description of our dataset. We have added descriptive information about the centre and the number of households. See lines 129-144 and the new table 1.

- Since only one case was confirmed by PCR, it could be useful if the authors reinforce the criteria to consider one case as Norovirus positive, maybe others clinical symptoms and signs with any epidemiological relationship. It will be informative if they also add who (laboratory) and how (methodology) it was performed the confirmatory PCR.

We added “Samples from suspected norovirus cases were sent to the Public Health Virology laboratory, Forensic and Scientific Services. A modified real-time reverse transcription PCR capable of detecting and differentiating norovirus genogroups GI and GII was utilised based on published primers and probes targeting ORF1-ORF2 junction region (Reference below). Sensitivity and specificity by in-house validation were deemed 100% and 99.06% respectively.”

Also, we defined a Norovirus outbreak as two or more similar illnesses resulting from a common exposure that is either suspected or laboratory-confirmed to be caused by norovirus (CDC definitions). See lines 68-77.

2. In the Discussion section:

- I consider that the authors could improve the discussion if they compare as much as the literature it allowed, to any similar study, even not necessarily Australian.

There are no detailed childcare centre norovirus outbreaks to compare with, other than large scale analysis done by Steele, which is discussed within. This is what prompted this study. See lines 49-54.

Minor points

1. Please delete the repeated word “that” in line 29.

Done

2. Please insert the reference in line 78.

There is no reference that we could find to reference this model. Its more of a statement describing the limits to our infectious disease or outbreak model.

3. In Figure 2, I advise to change the colors corresponding to unaffected and child care center circles to avoid any confusion with Figure 1 colors. I also recommend to improve the quality of the image, please consider to reproduce with another tools since it could be possible that the program not allowed.

We are not entirely sure what the issue was, as in all figures the red corresponds to the infant/child cases that bring the infection to the home cluster. The blue corresponds to infections at home. We have updated the figure.

4. I consider that it will be informative if the authors declare what they consider as children and infant, maybe in terms of age.

This suggestion adds value to the manuscript.See lines 84-87.

5. In my opinion, It will be more informative if the authors apply the primary or index case concept to the first case and secondary cases to the subsequently cases, and so on.

This has been changed throughout the manuscript.

Reviewer #2: Review of PONE-D-21-06174: "Extent of a norovirus gastroenteritis outbreak in a childcare centre: a household-level review," by Smoll et al.

Reviewer comments (by section of manuscript):

General: PLOS ONE "does not copyedit accepted manuscripts," and so this reviewer has endeavored to identify text that might be altered in the interest of being "clear, correct, and unambiguous."

Title: Consider adding "presumed" or "probable" before "norovirus."

We have discussed this at length within our group. Lines 64-83 provide some further definitions, and why we believe that this is a confirmed norovirus outbreak.

Abstract:

Several of this reviewer's comments on the body of the paper (see below) should be incorporated into the abstract. In particular, the distinction between norovirus and the broader category of gastroenteritis is quite important. Also, the term "affected persons" probably means "symptomatic household contacts of sick childcare attendees (and/or staff???)" but readers who have not yet read the full paper may not not interpret the term "affected" in that way. In the summary statements, it might be clearer to say that "in the study setting, each symptomatic childcare attendee apparently infected approximately two household contacts, on average."

This was felt to be an insightful comment, and improves the clarity of this manuscript. The abstract and the conclusion were altered to reflect these suggestions, and the actual sentence suggested was included. Thankyou.

Introduction:

This paper apparently describes the first investigation ever conducted of a presumed norovirus outbreak in childcare centers in Australia. Thus, it is of considerable public-health importance in the region, and may help set the standard for future surveillance.

Early in this section, it would be important to discuss the distinction between acute gastroenteritis and presumed or confirmed norovirus. Here or in "materials and methods" one could cite standard case definitions for "presumed norovirus," e.g. the Kaplan and/or Lively criteria (Kaplan J E, Ann Int Med 1982 June;96(6 pt 1):756-61. Lively J Y, Open Forum Infection Diseases 5(4): April 2018.).

These suggestions were incorporated into the methods. And added clarification about the distinction between presumed and confirmed in the introduction and methods. See Lines 64-83.

The statement that there is only "minimal information available regarding the R0 in childcare facilities" may only refer to the present study setting. For example, Steele et al. (Steele, MK et al. Characterizing norovirus transmission from outbreak data, United States. Emerging Infectious Diseases 2020; 26(8):1818-1825) evaluated norovirus data from 272 childcare facilities in the United States.

Excellent suggestion and has been added to the manuscript.

The basic biologic description of norovirus might be shortened.

We actually removed it as it does not add to the manuscript. Thanks for suggesting this.

To smooth the transition to the "materials and methods" section, the introduction might include a basic statement about the outbreak that was investigated - e.g. "An outbreak of suspected norovirus infection occurred in an Australian childcare center in August and September of 2020. We took advantage of this outbreak to provide a view......."

Excellent suggestion and incorporated this almost verbatim into the manuscript.

Materials and Methods:

In the first paragraph of the section, the paper offers a suspected case definition for gastroenteritis but none for norovirus. This is confusing, because the topic of interest (based on the paper's title) is norovirus, which must be distinguished from the broader category of "gastroenteritis." A case definition of "presumed norovirus infection" is needed, as noted above (criteria of Kaplan or Lively). A definition of "confirmed" norovirus is given, appropriately, based on fecal PCR testing.

Case and outbreak definitions were altered to reflect this, and the definition of Kaplan was used. As that is what is used in the Australian guidelines on Norovirus.

There should be a statement about the likely completeness of case ascertainment; for example, might children with only mild symptoms have been missed?

Excellent suggestion, we added to the limitations: “While the case interview method is good for identifying household cases, we did not call all families of the centre and there may be others that were unwell but did not report to the centre. Children with mild, or subclinical symptoms may also have been missed.”

Was there a specific procedure for identification of "parents of suspected cases?" And was there a standard interview guide for these parents? In particular, it would be helpful to know whether the interview was designed to distinguish systematically between probable norovirus and other types of gastrointestinal complaints.

We incorporated the following into the text: “The child care centre manager provides daily lists of suspected cases that are not attending or have been sent home due to symptoms of gastrointestinal illness. Information on family members affected, their symptoms and date of symptom onset, household size and likely route of transmission was collected, and ensured that other gastrointestinal complaints were not the cause of the symptoms.”

With regard to the identification of "rational transmission route," the concern about time course is necessary. But it would also be important to confirm that both childcare attendees and other household members had symptoms that met the norovirus case definition.

All household members had to meet case definitions. Added “…(must meet case definitions)” within a sentence.

Here or in the results section, it would be informative to state the number of stool specimens tested, the lag time between symptom onset and stool testing, and who was tested (symptomatic children, staff, household contacts; others). If many stool specimens were collected but failed to detect norovirus, the likely reasons for this pattern should be mentioned in the discussion section.

We had no negative stool specimens. The in-depth investigation began early and we scrambled to get stool specimens, and the parents were rather uncooperative and did not really see the need. “We were only able to obtain a single stool specimen and confirm a single case with PCR”

With regard to estimation of R0: Were all child-care staff, attendees, and household members presumed to be susceptible to norovirus at the outset of the outbreak?

We clarified this point: Our approach to modelling the basic reproduction number for this outbreak was performed using a time-dependent estimation of R0, and all childcare staff, attendees and household members assumed to be susceptible to norovirus.

Lines 74 - 79: The term "infector/infectee pair" doesn't seem quite right, given that several sick children were linked to multiple infectees (up to 3 in a single household per Figure 2). Perhaps "infector/infectee cluster" would work.

Infector/infectee pairs clusters were created by identifying primary cases that frequented the early care center and introduced the infection to the family cluster. The child that brought the infection home was paired with each family member to identify infector/infectee pairs used for the calculation of the serial interval.

Lines 80-81 (and 92): More justification should be given for using the denominator of “overall average attendees” instead of “total number of potentially exposed attendees.” Based on the data given, it seems that children who attended for some but not all days during the high-transmission interval might still have been at considerable risk of infection. Also, children who did not attend the childcare center every day probably do not comprise a random sample of the other attendees and might be systematically different in ways that could affect risk of acquisition of norovirus. Finally, might some of the absent children have stayed home because of norovirus symptoms?

It is difficult to assert the non-random mixing of children within the childcare center. We added this phrase: “The mixing may not have been entirely random as most, but not all children attended the center every day (is a limitation of our childcare center attendee attack rate).” All children that stayed home were alerted to us and a case history performed.

In lines 86 - 87: See comments re Figure 1.

Results:

A basic description of the study population might aid the reader. This could be presented in table form. It might describe total numbers of potentially exposed persons, not just numbers of persons who were known to be symptomatic, and categorize them by age (adult vs. child), childcare center vs. household status, and symptom status (in particular, whether symptoms satisfied the definition of presumptive norovirus). This suggestion is made because, at present, some information in the text is challenging to follow. For example, the reader is told that 47 persons in 18 households were "affected" but the text does not state that the 47 persons were apparently a mixture of childcare attendees and others, and the reader is not given the total population at risk.

The table helps to clarify things. Great suggestion.

A simple chronological description of the outbreak might also help the reader follow the narrative more easily. It could begin by stating - as the abstract does - that one childcare attendee had an episode of vomiting in August 2020 (was it August 24th?); that between 8/24/2020 and 9/18/2020, _____(number/%) of the attendees developed gastroenteritis symptoms and were sent home; and that on 8/31, the authors were informed of the outbreak, and efforts to control the outbreak were begun. This information already appears elsewhere but it would be useful to reiterate it in this section.

Great suggestion, the beginning of the results section was reworded.

Lines 92-102: This paragraph is hard to follow. For example, the relationships between the “average of 68 children” (line 92), the 3 infants and 22 children (line 95), the 16 children in line 96, and the 2 infants of line 97 are unlikely to be clear to the reader at first glance, and most of these numbers seem to denote numerators (of cases) but not denominators (of exposed population). In the same section, it might be wise to say "developed symptoms that met the case definition for presumed norovirus," instead of “were affected.”

These changes have been made and are improve the quality of the manuscript. The addition of the table helps to clarify denominators and numerators.

Lines 96-98: Were the two hospitalized infants a subset of the sixteen symptomatic children? Or were there eighteen symptomatic children?

This was clarified by modifying the following sentence: Two of the 16 children, both infant(s) were hospitalized for observation only.

Lines 104-111: Is this section meant to be the legend for Figure 1? This text might flow better as a figure legend than as a paragraph in the body of the paper. Similar query re lines 113-120.

This is a figure legend and formatted in the way suggested by Plos One

Line 123: “the index” vs. “the index child”

This has been rectified

Line 124: “they mixed” vs. “this child mixed”

This has been rectified

Line 127: Important to clarify the identity and subgroup composition of the 67 persons. If attack rate for households was 82.4% of 67 persons, total number of symptomatic persons should have been 55 (67 x 0.824). But per Figure 2, the total number of symptomatic childhood attendees and symptomatic household residents was only 44 (18 primary cases, 26 secondary cases), which would be 65.7% of the 67. Does the apparent discrepancy have something to do with the “excluded” cluster or clusters?

This was tricky, and we managed to clarify this, and make the story make more sense. 69 total persons in the household, minus 18 cases from the centre (two staff members included), for a total of 51 exposed persons, but only 23 transmissions occurred at home. See table 1 as well as updates in the results text, but the numbers have been reconciled.

Discussion:

Line 142 and subsequent paragraphs: See above re gastroenteritis vs. presumed or confirmed norovirus. The discussion should give the reader a data-based impression of the likelihood that this outbreak was caused by norovirus. The authors are entirely correct that norovirus is high on the list of etiologic possibilities, but the paper’s conclusion should be substantiated.

We had limited sample availability for this outbreak investigation, however large changes have been made to the manuscript to really remind the reader that this is probable norovirus.

Lines 144-5. See above re “attack rate of 82.4%.”

This has been rectified. 

Lines 147-155: Can you describe the Covid-19 mitigation strategies that were already in use at the childcare center before the gastroenteritis struck, especially those that might have had an impact on norovirus transmission? Also consider introducing the Covid-19 theme earlier in the paper as well as here.

See lines 62-63

Lines 149-152: Seems to conflict with previous observation that small household sizes limited possible magnitude of transmission. It would also be interesting (perhaps in future studies) to describe household-level strategies that might have prevented transmission, given that multiple households apparently had zero symptomatic transmission.

Lines 156-162: Very, very helpful and important.

Lines 163 – 170: Also very helpful and important. Some of this information might fit better in the introductory sections of the paper, though.

Lines 171-2: Again, more information is needed re number of stool specimens collected and possible explanations for absence of more confirmed norovirus specimens.

This has been rectified. “We were only able to obtain a single stool specimen and confirm a single case with PCR as parents often found it a hassle to obtain a specimen as the disease was transient and in some cases mild.“

Conclusion:

Lines 179-181: If the focus of the work really is on norovirus, and if the sick children and their sick household contacts had symptoms that satisfied the definition of presumed norovirus, it might be preferable to state that “in the outbreak under investigation, each child with presumed norovirus probably infected at least two other household members (on average).” However, based on what has been presented in the paper, it seems premature to make predictions about the generalizability of this estimate of R0 to other sites with different household compositions and/or different infection-control strategies (for example, if infection-control strategies had not been implemented when the outbreak investigation began, or if household sizes had been larger, far more people might have become symptomatic and the Ro would likely have risen).

This was a thoughtful comment, and we have added a comment about the generalizability of these findings early in the manuscript.

Lines 181-4: These recommendations and conclusions seem sound.

Figure 1:

This graphic gives the reader a very clear view of the evolution of the outbreak. It may also be the only mention of the very low number of infected childcare center staff. One drawback is that some of the red boxes denote children who attend the childcare center but other identical red boxes denote children who apparently did not attend; this is somewhat confusing but could be fixed easily.

It would be very helpful to see additional R0 estimates for the 1st week (Aug 24-31). Based on the visual evidence (17 cases in 7 days per the invaluable Figure 1), and the serial interval of approximately 2.5, R0 for the first week should be substantially higher than for subsequent weeks (after mitigation began).

The R0 for this outbreak is derived using the time-dependant method, and we use the first 7 days to provide the estimate. Essentially, it is impossible to provide an R0 for the first couple days as transmission has not occurred. We have added the calculation of R0 using different methodologies. 

Figure 2:

This is also a very appealing graphic, and it is fascinating to see the heterogeneity of apparent infectivity across households -- no secondary transmission at all in about 4 households, but 100% transmission in 4 other households; the reader may be quite interested to know if this is just chance or if something else is in play.

We think it important to highlight this variability that we cannot explain as yet. “The cause of the high variability of individual household attack rates (e.g. some with no transmission and some with 100% transmission) remains unclear.” See lines 174-181.

That said, Figure 2 might benefit from two modifications. First, the lines connecting different infection pathways sometimes overlap; the overlaps affect five households. The pathways for different households should be separated visually. Second, it would be helpful to the reader if the "unaffected" persons were described more fully. For example, are all the "unaffected" persons adult members of an affected child's household, or are some of them asymptomatic childcare attendees?

These comments are extremely helpful, and appreciate your suggestions. The figure and figure legend has been updated to match these suggestions.

---

## [Editor Report · Decision Letter 1]

1 Sep 2021

PONE-D-21-06174R1

A norovirus gastroenteritis outbreak in an Australian child-care center: a household-level analysis

PLOS ONE

Dear Dr. Smoll,

Thank you for submitting your manuscript to PLOS ONE. After careful consideration, we feel that it has merit but does not fully meet PLOS ONE’s publication criteria as it currently stands. Therefore, we invite you to submit a revised version of the manuscript that addresses the points raised during the review process.

The Authors have improved the manuscript substantially and addressed the reviewers’ comments. I have a remaining comment, 1. “The two generation model”/” The two generation network model” is not a standard model. I suggest the authors to remove the term in the main text and Figure 2 caption but describe the use of infector/infectee pairs (two generation) for the analysis as is (as have already done).

We look forward to receiving your revised manuscript.

Kind regards,

Eric HY Lau, Ph.D.

Academic Editor

PLOS ONE

Journal Requirements:

Additional Editor Comments (if provided):

The Authors have improved the manuscript substantially and addressed the reviewers’ comments. I have a remaining comment,

1. “The two generation model”/” The two generation network model” is not a standard model. I suggest the authors to remove the term in the main text and Figure 2 caption but describe the use of infector/infectee pairs (two generation) for the analysis as is (as have already done).
---

## [Author Response · Author response to Decision Letter 1]

2 Sep 2021

We agree with the Editors recommendation and have made the changes in the 3 areas of the manuscript.

Thank you

---

## [Editor Report · Decision Letter 2]

14 Oct 2021

A norovirus gastroenteritis outbreak in an Australian child-care center: a household-level analysis

PONE-D-21-06174R2

Dear Dr. Smoll,

We’re pleased to inform you that your manuscript has been judged scientifically suitable for publication and will be formally accepted for publication once it meets all outstanding technical requirements.

Kind regards,

Eric HY Lau, Ph.D.

Academic Editor

PLOS ONE
---

## [Editor Report · Acceptance letter]

25 Oct 2021

PONE-D-21-06174R2 

A norovirus gastroenteritis outbreak in an Australian child-care center: a household-level analysis 

Dear Dr. Smoll:

I'm pleased to inform you that your manuscript has been deemed suitable for publication in PLOS ONE. Congratulations! Your manuscript is now with our production department. 

Kind regards, 

on behalf of

Dr. Eric HY Lau 

Academic Editor

PLOS ONE